# A Non-Resonant Piezoelectric–Electromagnetic–Triboelectric Hybrid Energy Harvester for Low-Frequency Human Motions

**DOI:** 10.3390/nano12071168

**Published:** 2022-03-31

**Authors:** Gang Tang, Zhen Wang, Xin Hu, Shaojie Wu, Bin Xu, Zhibiao Li, Xiaoxiao Yan, Fang Xu, Dandan Yuan, Peisheng Li, Qiongfeng Shi, Chengkuo Lee

**Affiliations:** 1Jiangxi Province Key Laboratory of Precision Drive & Control, Nanchang Institute of Technology, Nanchang 330099, China; tanggang@nit.edu.cn (G.T.); 15797625058@163.com (Z.W.); xinh9602@163.com (X.H.); wu2020313111@163.com (S.W.); 2010994268@nit.edu.cn (B.X.); lizhibiao@nit.edu.cn (Z.L.); qwwxiaoy@163.com (X.Y.); 2021994767@nit.edu.cn (F.X.); yuandandan@nit.edu.cn (D.Y.); lipeisheng@ncu.edu.cn (P.L.); 2Department of Electrical & Computer Engineering, National University of Singapore, 4 Engineering Drive 3, Singapore 117583, Singapore; 3School of Electronic Science and Engineering, Southeast University, Nanjing 210096, China

**Keywords:** triboelectric, piezoelectric, electromagnetic, self-powered, hybrid

## Abstract

With the rapid development of wireless communication and micro-power technologies, smart wearable devices with various functionalities appear more and more in our daily lives. Nevertheless, they normally possess short battery life and need to be recharged with external power sources with a long charging time, which seriously affects the user experience. To help extend the battery life or even replace it, a non-resonant piezoelectric–electromagnetic–triboelectric hybrid energy harvester is presented to effectively harvest energy from low-frequency human motions. In the designed structure, a moving magnet is used to simultaneously excite the three integrated energy collection units (i.e., piezoelectric, electromagnetic, and triboelectric) with a synergistic effect, such that the overall output power and energy-harvesting efficiency of the hybrid device can be greatly improved under various excitations. The experimental results show that with a vibration frequency of 4 Hz and a displacement of 200 mm, the hybrid energy harvester obtains a maximum output power of 26.17 mW at 70 kΩ for one piezoelectric generator (PEG) unit, 87.1 mW at 500 Ω for one electromagnetic generator (EMG) unit, and 63 μW at 140 MΩ for one triboelectric nanogenerator (TENG) unit, respectively. Then, the generated outputs are adopted for capacitor charging, which reveals that the performance of the three-unit integration is remarkably stronger than that of individual units. Finally, the practical energy-harvesting experiments conducted on various body parts such as wrist, calf, hand, and waist indicate that the proposed hybrid energy harvester has promising application potential in constructing a self-powered wearable system as the sustainable power source.

## 1. Introduction

In recent years, with the advancement of sensing and fabrication technologies, numerous micro-power Internet of Things (IoT) sensors and electronic devices have been developed rapidly [1,2], but how to achieve sustainable energy supply to such a large number of devices remains a grand challenge. The bulky size, limited service life, and environmental pollution issue of traditional chemical batteries have gradually become the bottlenecks restricting the further development of micro-power devices. In this regard, scavenging ambient energies such as light, heat, wind, and mechanical vibration using energy harvesters appears to be a promising alternative [3,4]. At present, researchers across the world have performed detailed investigations of using energy harvesters to power wearable devices. In 2011, Yun et al., proposed a self-powered wireless sensor node based on solar energy harvesting, and the output power can reach 20 mW under outdoor conditions [5]. Although the output of solar energy harvesting is relatively large, it is also easily affected by the environmental parameters, leading to no electrical output in the absence of light. In addition, another form of energy—thermal energy—has also been studied by researchers for energy harvesting. For example, Wang et al., connected 4700 thermocouples in series and attached them to the human body, which can produce 0.15 V open-circuit voltage and 0.3 nW output power at room temperature [6]. Although the collection of thermal energy could be weather independent, the output power is too low due to the limited temperature difference between the human body and the environment, which is normally insufficient to power wearable devices. The above-discussed two mechanisms exhibit high dependence on the ambient conditions and could be greatly influenced by external factors; thus, they are not ideal as a steady power supply for wearable devices. In this regard, a lot of researchers have turned to use mechanical vibration energy, which is more ubiquitous in the environment and human activities for electrical energy generation. This could be an ideal approach, since it is not easily interfered with by other external factors such as weather and temperature. Thus, it exhibits a broad application prospect in supplying energy for wearable and/or microscale electronics.

To effectively harvest the mechanical energy existing in the base excitations, biological motions, and flow waves [7,8], a large variety of vibrational energy harvesters have been proposed based on different mechanisms, e.g., piezoelectric [9,10,11,12,13,14], electromagnetic [15,16,17,18,19], electrostatic [20,21], triboelectric [22,23,24,25,26], and magnetostriction energy harvesters [27]. Moreover, two or more mechanisms can also be synergistically combined, which is called a hybrid mechanism [28,29,30]. With optimized design, the integrated mechanisms can work synergistically and complementarily, leading to an improvement in the output performance, transducing efficiency, and device’s adaptability [31,32]. To realize a hybrid mechanism, the triboelectric mechanism is usually adopted to be integrated with other mechanisms due to its great advantages of simple structure, easy fabrication, broad materials choice, diverse operation modes, low cost, and high scalability [33,34,35]. In 2012, the first triboelectric nanogenerator (TENG) was invented by Wang’s team based on the coupling of contact electrification and electrostatic induction between two different materials [36]. After that, TENG has received tremendous research efforts and has been proven as a promising technology for energy harvesting and self-powered sensing [37,38,39,40,41], for the application scenarios involved with body motions [42,43], wind [44,45], water waves [46,47], vibrations [48,49], rotations [50,51], and even acoustic waves [52,53].

In 2015, Li et al., designed a triboelectric–piezoelectric composite energy harvester on clothes containing several carbon fibers for human energy harvesting, with zinc oxide nanorods as the piezoelectric material and polydimethylsiloxane (PDMS) as the triboelectric material [54]. In 2020, Rahman et al., reported a miniaturized freestanding hybrid generator based on TENG and an electromagnetic generator for harvesting the human vibrational energy during motions [55]. Then, Li et al., proposed a flexible motion monitoring device based on a magnetic microneedle array [56]. The magnetic microneedles in this structure not only act as the curved magnetic poles of the electromagnetic generator but also act as a triboelectric friction layer. The closed and curved characteristics of the microneedles are used to complete the energy-harvesting action. These examples have proven that the hybrid strategy is feasible and effective for output performance and efficiency enhancement. Yet, most of them only utilize two mechanisms, and a more efficient way to harvest the low-frequency human motions is still highly desirable.

In order to harvest the mechanical energy from human motions more efficiently and improve the output performance, the synergistic effect between the adopted mechanisms in the hybrid energy harvester should be carefully designed. Generally speaking, piezoelectric and triboelectric energy harvesters can generate large output voltage with large internal impedance, and electromagnetic energy harvesters can produce large output current with low internal impedance. Therefore, these three energy-harvesting mechanisms can complement each other and improve the overall output performance of a hybrid device. Herein, we propose a novel non-resonant hybrid energy harvester with the piezoelectric–electromagnetic–triboelectric mechanism for effectively harvesting the low-frequency energy from various human activities. The structure of the device is configured into a cylindrical shape, which is convenient to be held in the hand or fixed to a certain part of the body. When the human body moves, the Al-coated magnet in the cylindrical structure starts to reciprocate inside the cavity, inducing power generation on the polyvinylidene fluoride (PVDF) piezoelectric elements at the two ends, the triboelectric element on the cavity sidewall, as well as the electromagnetic element winding around the middle part simultaneously. Under the conditions of an excitation frequency of 4 Hz and a movement displacement of 200 mm, each mechanism achieves the following output voltage and current: 9.4 V and 21.4 mA for the electromagnetic generator (EMG); 95 V and 106 μA for the piezoelectric generator (PEG); and 150 V and 1.4 μA for the TENG. Then, the generated outputs are regulated through an energy-harvesting circuit and finally stored in a capacitor for powering IoT sensors. The integrated outputs from the hybrid energy harvester can charge a 2.2 μF capacitor up to 13 V in only 1 s. Therefore, this hybrid structure proposed in this work shows great potential as a sustainable power source for various IoT sensor nodes and wearable devices in the IoT era.

## 2. Results and Discussion

### 2.1. Device Structure

Figure 1a shows the 3D schematic view of the hybrid energy harvester, i.e., a hybrid generator, which is mainly a symmetrical cylinder structure. Two lightweight and hollow half-cylinders with a dimension of 117 mm × 36 mm (length × diameter) are prepared by 3D printing and used as the supporting frame. Two caps with whorls are also fabricated for covering both ends of the tunnel with the same 3D printing method. Overall, the hybrid generator consists of one EMG at the middle, two PEGs at the two ends, and one TENG along the inner surface. A cylindrical NdFeB permanent magnet (with a diameter of 10 mm and height of 30 mm) is adopted as the proof mass to respond to environmental perturbations and trigger all three energy-harvesting units with synergistic output generation. To construct the EMG unit, the middle part of the hollow cylinder is wound with copper (Cu) wires (with a diameter of 70 µm) for 1000 turns, forming the coil structure to capture the magnetic flux variation. In terms of the PEG unit, two commercially available PVDF thin films with high piezoelectric coefficient and high durability are used. In order to make use of the large mechanical impact from the magnet more efficiently, PVDF films are attached to the top and bottom of the cylinder frame with both ends fixed, forming the suspended membrane configuration that is more sensitive to impact force. The PVDF film is coated with silver on both sides as upper and lower electrodes, which is further encapsulated with polymer films for insulation and protection. When subjected to external vibration, the cylindrical magnet oscillates and impacts the PVDF films, causing structural deformation, thereby generating piezoelectric outputs accordingly. To construct the TENG unit, two aluminum (Al) electrodes are first attached to the upper part and the lower part of the hollow cylinder, which is followed by the attachment of a poly tetra fluoroethylene (PTFE) thin film as the negative triboelectric material. To improve the triboelectric output performance by the magnet friction, a more positive Al foil is attached to the surface of the magnet, acting as the positive triboelectric material. Figure 1 shows a digital photograph of the assembled device, while its diameter (≈36 mm) and length (≈117 mm) are indicated in Figure 1e,f, respectively.

### 2.2. Working Principle

According to the structural design of the hybrid generator, synergistic outputs could be generated by the three integrated units: EMG, PEG, and TENG. The EMG unit uses the movement of the magnet to change the magnetic flux of the Cu coil and generates current based on Faraday’s law of electromagnetic induction. When the permanent magnet moves to the ends and impacts the PVDF films, a piezoelectric current is generated from the PEG unit based on the piezoelectric effect. Meanwhile, the triboelectric charges on the magnet surface induce potential difference on the two triboelectric electrodes during this reciprocating movement, leading to current generation on the TENG unit. All these three units are triggered by the magnet movement simultaneously, generating synchronized outputs that can be further regulated to increase the overall output power of the device.

As shown in Figure 2, the detailed working principle of the hybrid generator is illustrated. When the overall structure of the device is not subject to external vibration, it is in an equilibrium state and has no output. As shown in Figure 2a, when the device is shaken to the right, the permanent magnet hits the PVDF film on the left end due to inertia, which deforms the PVDF film and generates a corresponding piezoelectric current. Meanwhile, due to the electrostatic equilibrium of triboelectric charges, there is no current generated from the two Al electrodes. Similarly, the magnetic flux through the coil does not change, and there is no corresponding electromotive force and induction current in the Cu coil. When the external force changes direction, the magnet separates from the PVDF film at the left end and moves toward the right end, as shown in Figure 2b. Upon separation, a small piezoelectric current is generated when the PVDF film restore to its original state. Then, during this movement of the magnet, the magnetic flux of the Cu coil changes significantly. According to Faraday’s law of electromagnetic induction, an electromotive force is generated, giving rise to an induction current in the Cu coil. In the TENG unit, the movement of the magnet breaks the previous electrostatic balance and induces potential difference on the two Al electrodes. Thus, the current flows from the high potential end (right) to the low potential end (left), generating a corresponding triboelectric output. When the magnet continues to move to the right, it finally hits the other PVDF film, as shown in Figure 2c. At this time, a corresponding piezoelectric current is generated, and the outputs of the EMG unit and the TENG unit come to rest due to electromagnetic and electrostatic balance. Then, with the applied force changing direction again, the magnet will move away from the right PVDF film toward the left end, as shown in Figure 2d. Similar to the process in Figure 2b, a small piezoelectric output current is first generated, which is followed by the generation of EMG and TENG outputs in a reverse direction, until a new equilibrium is achieved. After that, another new cycle of output generation starts. With the reciprocating movement of external force, this output generation process from these three units will repeat and continue.

### 2.3. Output Characterization

In this section, the output performance of the developed piezoelectric–electromagnetic–triboelectric hybrid generator is analyzed quantitatively. When the magnet moves, all the three generator units can generate synchronized outputs, which will be investigated one by one. As shown in Figure 3, with a vibration frequency of 4 Hz and a vibration displacement of 200 mm, the output performance of the TENG unit is measured when using magnets with different lengths.

Under this excitation condition, it can be seen from Figure 3a that the triboelectric output voltage can reach 125 V. In order to further investigate the output characteristics of the TENG unit during this process, the output voltage in one cycle is analyzed. It can be divided into the following five stages. (1) The t_1_ stage is the initial stage. At this stage, the movement direction of the magnet is vertically upward, while the upper end of the magnet is still within the lower triboelectric electrode range. Thus, there is no charge transfer between the two triboelectric electrodes due to electrostatic balance, leading to zero output voltage. (2) In the t_2_ stage, the magnet continues to move upward, beginning to cross the electrode boundary and enter the upper electrode range. Hence, charge transfer occurs due to the arising potential difference, and a positive voltage signal is then produced. (3) At the t_3_ stage, the magnet fully enters the upper electrode range, reaches the top, and moves downward. Similar to the t_1_ stage, there is no charge transfer and output voltage before the magnet reaches the lower electrode range. (4) At the t_4_ stage, the magnet continues to move downward and begins to enter the lower electrode range. At this moment, charge transfer occurs in the opposite direction, producing a negative voltage signal. (5) During the t_5_ stage, the magnet fully enters the lower electrode range and returns to the original position as in the t_1_ stage, ready for another new cycle to start. The charge transfer and output generation process will continue if the excitation is repeated.

According to the output characteristics of TENG, the length of the magnet could be an important factor affecting the output performance; thus, it is investigated here. As shown in Figure 3b, it can be found that as the length of the magnet increases, the positive voltage peak first increases and then decreases, with a maximum value of 163 V achieved at 40 mm length. For the negative voltage peak, it gradually increases. The corresponding current outputs can be seen in Figure 3c. Based on the overall outputs of the voltage and current, the appropriate magnet size should be 40 mm. In order to further determine the appropriate magnet length, we introduce a new comparison parameter, i.e., Vpp/G (unit: V/N), which is the ratio of the peak-to-peak output voltage to the device weight in gravity. As shown in Figure 3d, the output voltage per unit weight of the TENG shows a trend of first increasing and then decreasing. Although the peak-to-peak output voltage is 288 V when the magnet length is 40 mm, the output voltage per unit weight of 30 mm is greater, 63.168 V/N. Since the device aims at wearable applications, a higher output voltage per unit weight will be more efficient in energy harvesting and more convenient for users, and thus, the most suitable length of the magnets for the TENG unit is 30 mm.

As shown in Figure 4, the output performance of the EMG unit is measured with the same vibration frequency of 4 Hz and displacement of 200 mm. Here, the influence of the length of the magnet on the output performance of the EMG unit of the device is also discussed. Figure 4a shows the output voltage of the EMG unit from 0.1 to 1.1 s, where the positive voltage is 9.4 V, and the negative voltage is −11 V. This asymmetric output is because in the adopted manual excitation, the magnet is accelerating from one end to the other, meaning that the approaching speed toward the Cu coil is less than the leaving speed. Thereby, the negative peak due to the magnet leaving is relatively larger than the positive peak due to the magnet approaching.

To further study the output characteristics of the EMG unit, the output waveforms are investigated in detail. In the magnet movement process from one end to the other, the output signal of the EMG unit can be divided into the following four stages. (1) In the t_1_ stage, the magnet has just started to move and is far from the position of the coil, and thus, there is no signal output due to the barely changing magnetic field. (2) During the t_2_ stage, the magnet is gradually approaching the coil, and the induced magnetic flux in the coil is increased, leading to the generation of the positive peak. (3) In the t_3_ stage, the magnetic in the coil reaches the largest, and the output current thus gradually comes to zero. After that, the magnet starts to leave the coil, resulting in a decrement in the magnetic flux. Then, an opposite output, i.e., the negative output peak, is generated. (4) At the t_4_ stage, the magnet is far from the coil again; thus, there is no signal output as well. When the magnet moves from right to left, a similar positive–negative output will be generated due to the same increment–decrement trend of the magnetic flux.

Under the premise of the same conditions, the change of the magnetic flux will be affected by the length of the magnet. Thus, the influence of different magnet lengths on the induced outputs of the EMG unit is studied. The output signals using different magnet lengths are measured at the same vibration frequency of 4 Hz and displacement of 200 mm, as shown in Figure 4b,c. It can be seen that for the positive voltages, it first increases slightly, reaching a maximum of 9.8 V at 30 mm, and then decreases. For the negative voltages, it reaches a maximum of 15 V at 40 mm. When the length is 20 mm or 30 mm, the body of the magnet at the ends is not overlapping with the coil position in the initial position (t_1_ stage). The main influencing factor here is the magnetic flux difference when the magnet is at the middle position and the initial position. Since the magnet is not overlapping with the coil in the initial position, the magnetic flux can be considered very small for both the 20 mm and 30 mm cases. Then, at the middle position, the 30 mm magnet produces a stronger magnetic flux, which leads to a higher output. When the length of the magnet is 40 mm, 50 mm, or 60 mm, the magnet has begun to overlap with the coil. The initial magnetic flux increases with the magnet length, and thus, the magnetic flux difference will be decreased. Moreover, the acceleration time of the longer magnet is also reduced, resulting in a smaller moving speed, so that the output will gradually decrease.

Similarly, the output voltage per unit weight is also used to evaluate the EMG’s working efficiency of magnets with different lengths. As indicated in Figure 4d, when the magnet length is 30 mm, the output voltage is the largest (22.4 V), followed by 40 mm. However, as far as Vpp/G is concerned, when the length of the magnet increases, the output voltage per unit weight shows a gradually decreasing trend. The maximum output voltage per unit weight is 7.718 V/N for the 20 mm magnet, which is followed by 5.611 V/N for the 30 mm magnet. Therefore, in terms of the EMG output, the optimal length is 20 mm, followed by 30 mm.

As shown in Figure 5, the output performance of the PEG unit is analyzed. As discussed above, to ensure that the TENG and EMG units have larger outputs, the magnet length is determined to be 30 mm. The voltage signal of the PEG unit measured under 4 Hz and 200 mm displacement is shown in Figure 5a, while Figure 5b shows the movement of the magnet in one signal generation period of the PVDF film. It can be divided into four stages. (1) During the t_1_–t_2_ stage, the magnet is not in contact with the PVDF film, and thus, no signal is generated. (2) In the t_2_–t_3_ stage, the magnet collides with the PVDF film and deforms it in a very short time, inducing a sharp positive output. (3) At the t_3_–t_4_ stage, the magnet is separating from the PVDF film, which gradually restores to its original state by its own elastic force, resulting in the generation of a small negative output. (4) During the t_4_–t_1_ stage, the magnet moves downward and starts to prepare to collide with the other PVDF film.

Through the above investigations, it can be concluded that the most suitable magnet length for the hybrid generator is 30 mm. Since the device is mainly used in light activities such as human walking, it is necessary to analyze the device’s output performance with varying frequencies and displacements. From Figure 6a,d,g, it can be observed that the outputs of the TENG, PEG, and EMG units all increase with the vibration frequency. This is because when the frequency increases, the period for charge transfer and output generation is shortened, leading to the increment of output currents and voltages. Then, as shown in Figure 6b,e,h, the influence of movement displacement on the output performance is investigated. A similar increment trend can be observed, which is because larger displacement induces a higher moving speed of the magnet. As a result, the output performance of each generator unit is increased.

Figure 6c,f,i show the output voltage and power curve for each generator unit when connecting to different external loads, with a magnet length of 30 mm, vibration frequency of 4 Hz, and displacement of 200 mm. The output power is calculated from the output voltage, using the following equation:(1)$$P\;=\;\frac{U_{P}^{2}}{R}$$

Among them, *R* refers to the resistance of the external load, *U_P_* refers to the output voltage on that load, and *P* is the corresponding output power. It can be seen that all the output voltage curves first increase continuously with the external resistance and finally stabilize. This phenomenon leads to a maximum output power when the internal resistance of the TENG, EMG, or PEG unit matches the external resistance. By comparison, the internal resistance of TENG is the largest, which is followed by the PEG, and finally the EMG. It is calculated that under this excitation condition, the hybrid generator obtains the maximum output power of 63 μW at 140 MΩ for one TENG unit, 87.1 mW at 500 Ω for one EMG unit, and 26.17 mW at 70 kΩ for one PEG unit.

### 2.4. Device Application

The hybrid generator is designed to harvest kinetic energy from the human body, so it is necessary to investigate its performance under various human activities. As shown in Figure 7, four positions at the wrist, hand, calf, and waist are selected to wear the hybrid generator for testing, following the exercise conditions: wrists and hands in situ swinging; calf in situ stepping; and waist in situ twisting. Under these conditions, the peak-to-peak output voltages at each position are summarized in Table 1. According to the results, the hybrid generator can effectively capture the kinetic energy from the body’s routine movements.

In practical applications, constant current and voltage from a power source are more usable for driving electrical sensors and other components. Therefore, the pulse-like outputs generated from the hybrid generator need to be regulated and stored in an energy storage unit first, such as a capacitor or chemical battery [57,58,59].

A customized circuit is designed for the hybrid generator. The outputs from each generator unit are first rectified, which are then connected in parallel to the inputs of a commercial power management chip (LTC3588). The outputs from the chip are further connected to a capacitor for energy storage, as shown in Figure 8a. The capacitor charging capability of each generator unit and their combination is compared using a 2.2 μF capacitor. As illustrated in Figure 8b, the charging speed of the three-unit combination is remarkably faster than that of a single unit. It indicates that the hybridization of multiple mechanisms is indeed a good strategy to improve the energy-harvesting efficiency. The voltage of the capacitor at 100 s when charging with the TENG unit is only 10.5 V, while 11–13 V can be achieved when other units or combination are used. This can be attributed to the fact that TENG normally has a large inner impedance and low output current, hindering its charging capability. In this regard, the EMG and PEG units with relatively large output current can be good complements to the TENG unit, enhancing the charging capability when configured in a hybrid generator.

After the power management circuit, the stored energy in the capacitor can be used to drive IoT sensors. Here, a wireless temperature/humidity sensor is used as an example to show the capability of the hybrid generator. As depicted in Figure 8c–e, the sensor is successfully powered by the outputs from the hybrid generator when it is subjected to a vibration frequency of 4 Hz. The sensor starts to work in about 10 s after the generator device is actuated, and it feeds back the temperature and humidity signal to the mobile phone interface through Bluetooth in a real-time and intermittent manner. The entire process can last for 18 s.

## 3. Conclusions

In this work, a piezoelectric–electromagnetic–triboelectric hybrid generator is developed to harvest energy from various human activities. Benefited by its facile structure, non-resonant nature, and broad responsiveness, the hybrid generator is able to perform effective energy harvesting even with low-frequency human motions. The hybrid generator has a small dimension of ⌀36 mm × 117 mm and a light weight, which is suitable to power intelligent mobile devices and wearable electronics. The design integrates three energy-harvesting mechanisms in a synergistic manner to achieve greater output performance. In addition, it also shows good adaptability for wearable applications; for example, it can be worn on a person’s wrist, calf, waist, etc., to effectively harvest the kinetic energy from human activities. The dynamic response and performance of each mechanism, as well as the optimal conditions such as magnet length, have been analyzed in detail. Furthermore, the output performance of capacitor charging has been measured for every single unit and three-unit combination. The results reveal that the charging performance of the unit combination is remarkably higher than that of an individual unit, showing the feasibility and advantages of using the hybrid design. Benefited by the superior output performance, the hybrid generator can drive a wireless IoT module in real time, which can collect both the temperature and humidity information and send it to a mobile phone via Bluetooth. The designed hybrid generator in this work demonstrates a promising strategy to integrate multiple working mechanisms into a hybridized structure, for more effective energy harvesting and self-powered monitoring in wearable and IoT applications.

## Figures and Tables

**Figure 1 nanomaterials-12-01168-f001:**
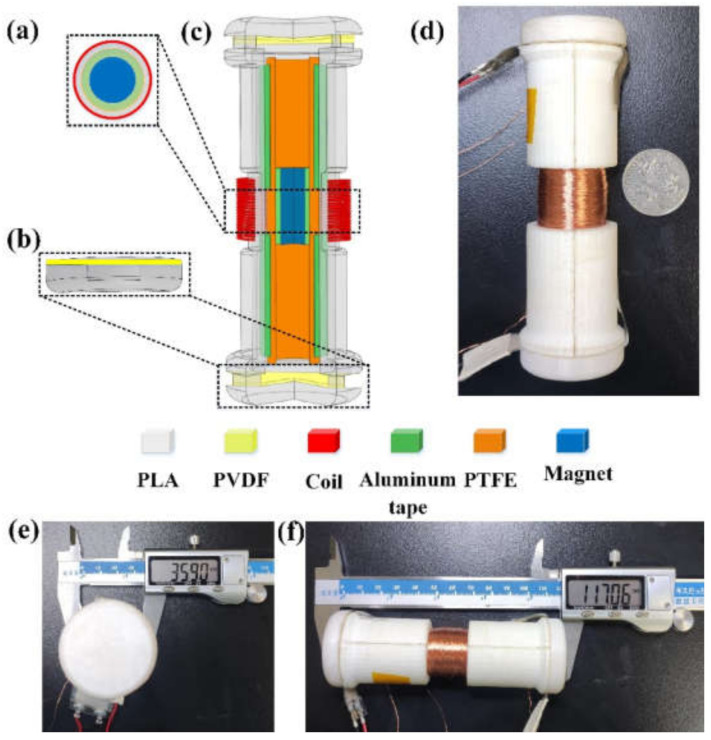
Structure of the piezoelectric–electromagnetic–triboelectric hybrid energy harvester. (**a**) TENG-EMG unit. (**b**) PEG unit simply supported beam structure. (**c**) Three-dimensional (3D) schematic diagram of the proposed hybrid energy harvester. (**d**) Photograph of the hybrid energy harvester. (**e**,**f**) The diameter and length of the hybrid energy harvester.

**Figure 2 nanomaterials-12-01168-f002:**
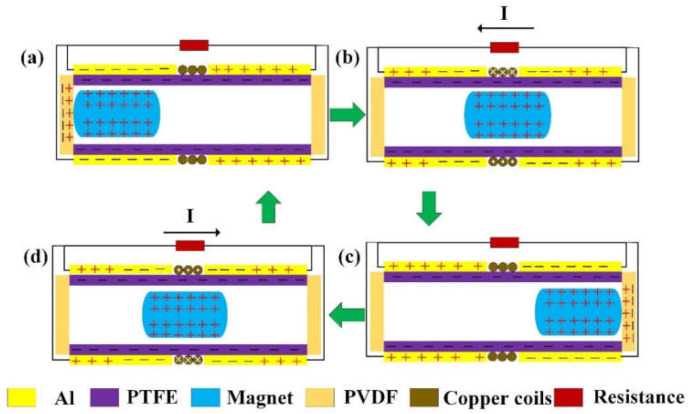
Schematic diagrams showing the operation mechanism and the corresponding current flow from the three generator units, i.e., EMG, PEG, and TENG. (**a**) The magnet on the left end and impacting the left PVDF film. (**b**) The magnet moving from left to right. (**c**) The magnet on the right end and impacting the right PVDF film. (**d**) The magnet moving from right to left.

**Figure 3 nanomaterials-12-01168-f003:**
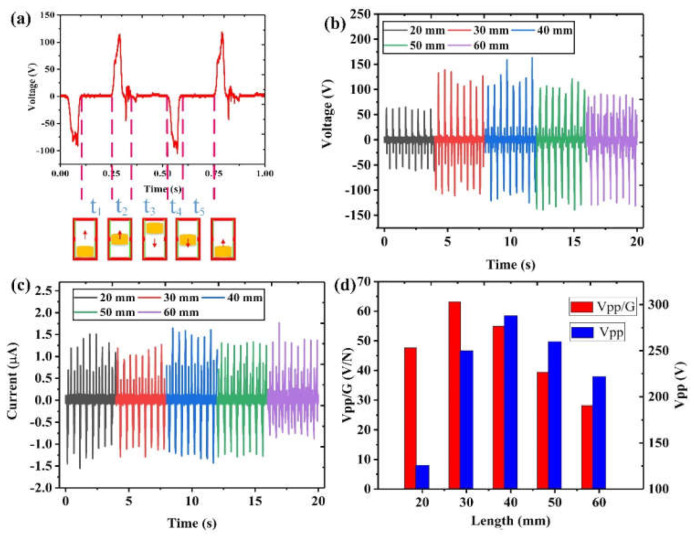
Output characteristics of the TENG unit. (**a**) Triboelectric output voltage corresponding to different stages within one cycle. (**b**) Output voltages and (**c**) output currents of the TENG unit under different magnet lengths. (**d**) Performance comparison of the output voltage and voltage per unit weight for different magnet lengths.

**Figure 4 nanomaterials-12-01168-f004:**
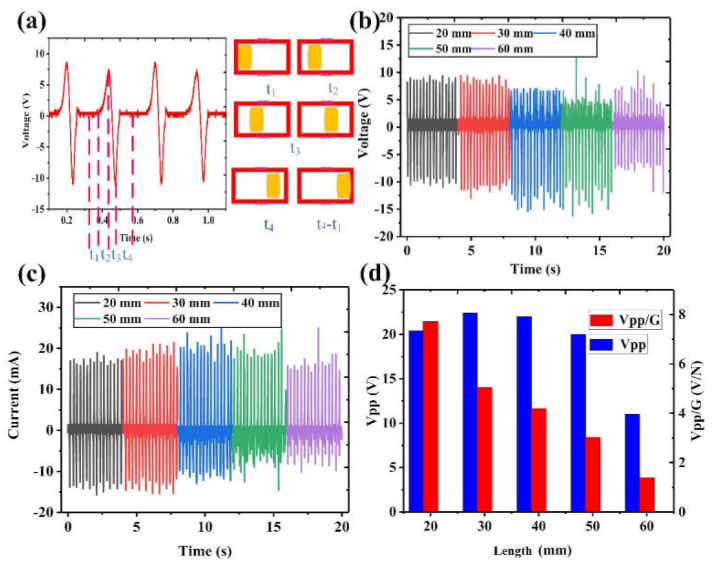
The output characteristics of the EMG unit. (**a**) The output voltage corresponding to different stages within one cycle. (**b**) Output voltages and (**c**) output currents of the EMG unit under different magnet lengths. (**d**) Performance comparison of the output voltage and voltage per unit weight for different magnet lengths.

**Figure 5 nanomaterials-12-01168-f005:**
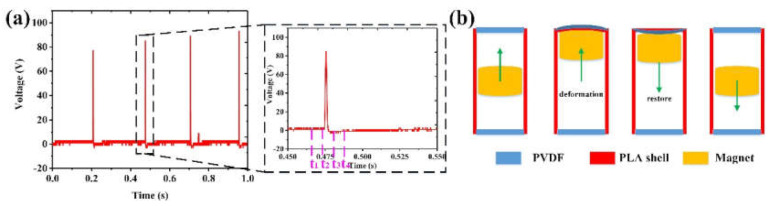
The output characteristics of the PEG unit. (**a**) The output voltage corresponding to different stages within one cycle. (**b**) Magnet movement and PVDF status in different stages.

**Figure 6 nanomaterials-12-01168-f006:**
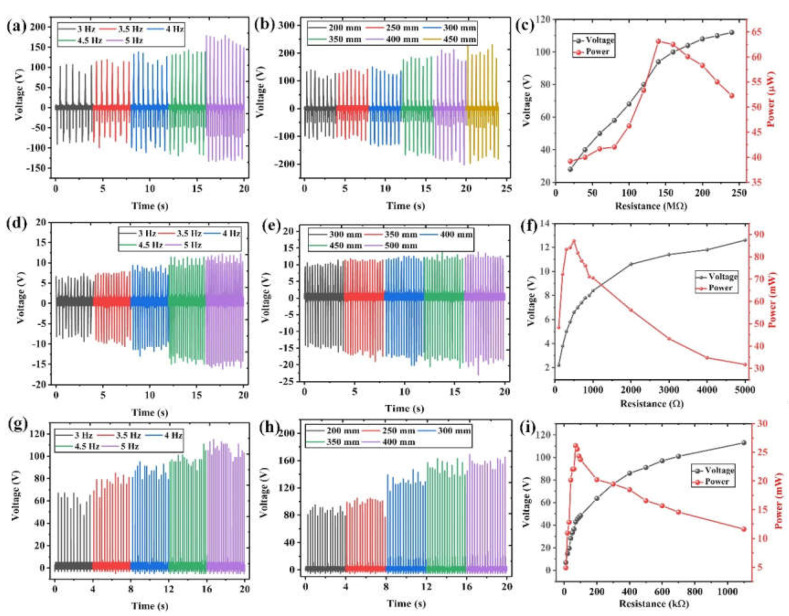
The output voltages of the TENG unit at (**a**) different frequencies and (**b**) different displacements, and (**c**) the output voltage and power curve under different external loads. The output voltages of the EMG unit at (**d**) different frequencies and (**e**) different displacements, and (**f**) the output voltage and power curve under different external loads. The output voltages of the PEG unit at (**g**) different frequencies and (**h**) different displacements, and (**i**) the output voltage and power curve under different external loads.

**Figure 7 nanomaterials-12-01168-f007:**
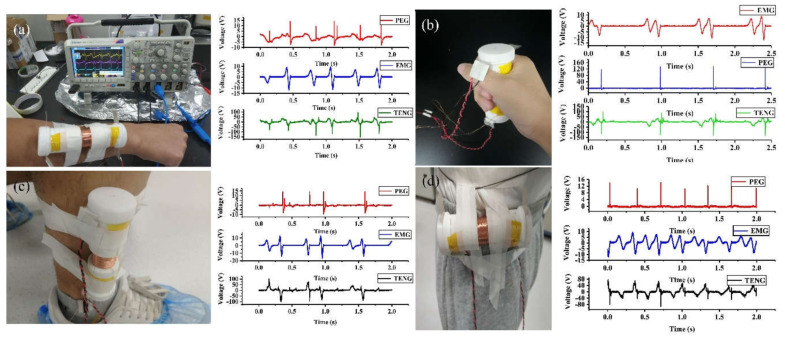
Practical implementation of the hybrid generator to harvest the human activity energy on various body parts: (**a**) wrist; (**b**) hand; (**c**) calf; (**d**) waist.

**Figure 8 nanomaterials-12-01168-f008:**
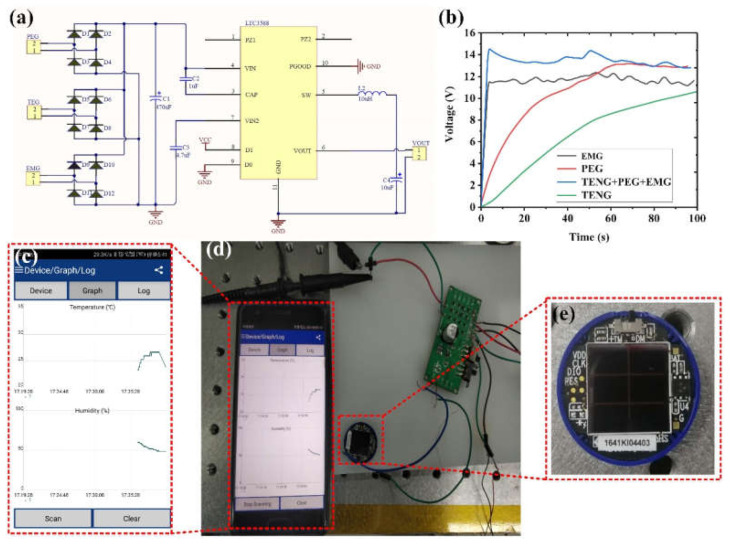
(**a**) Power management circuit for the hybrid generator. (**b**) Charging a capacitor of 2.2 μF. (**c**–**e**) Powering a wireless IoT module by the hybrid generator.

**Table 1 nanomaterials-12-01168-t001:** Output voltage generated from each wearing position (Vpp is the peak-to-peak voltage in volts).

Position	PEG (Vpp)	EMG (Vpp)	TENG (Vpp)
Wrist	21	22	180
Calf	20	24	200
Hand	120	15	220
Waist	14	17.5	140

## Data Availability

The data of our study are available upon request.

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
