# Peer review of "A Non-Resonant Piezoelectric–Electromagnetic–Triboelectric Hybrid Energy Harvester for Low-Frequency Human Motions"

_nanomaterials, 2022, doi:10.3390/nano12071168_

Round 1

Reviewer 1 Report

In this manuscript, the authors have prepared a hybrid piezoelectric-electromagnetic-triboelectric generator to harvest energy from the various part of the human body during physical activities. The device structure (design), working principle, and output characteristics, were demonstrated in the manuscript. Additional to the facial structure and non-resonant nature, the hybrid generator is able to effectively harvest the energy at low frequencies from the human body during motions as well as can be utilized for the wearable applications.

The overall evaluation of the current manuscript is good. However, the underlying mechanism and the experimental results of the hybrid energy harvester are little puzzling and unclear. The authors are required to provide further explanations with supporting experimental data. Unless all of the major/minor comments raised by the reviewer is addressed clearly with additional data, this manuscript should not be accepted to the publication in the Journal of Nanomaterials.

Major and minor comments

1) In the working principle of EMG, the main influencing factor is the varying magnetic flux, which is dependent on the length of the permanent magnet and number of coils (length of the number of coils). The authors stated about the length of the magnetic bars. However, the length of the coils is not clear; only they mentioned the numbers of turns are 1000. The authors need to put the exact length of the coil.

2) The authors have mentioned that the 30 mm longer magnetic bar produces a strong magnetic flux resulting into higher output voltage and current. While for higher lengths (40, 50, and 60 mm) the magnetic field overlapped leading to lower outcome of the voltage and current. This point need more clear explanation. Why the 30 mm magnetic produces a higher magnetic flux compare to others? The authors did not mentioned about the strength of the magnetic bars. Does the author have any data for the magnetic strength of the each magnetic bar?

3) Does the length of the magnetic bar is smaller, equal, or larger than length of the coil. More importantly, the authors need to compare the length of the coil and magnetic bar. It should be clarified that the overlapping of the magnetic field between the magnet and coil by comparing the length of the coil and magnetic bar (i.e., by taking the length of the magnetic bar smaller, comparable (same size), and longer than the length of the coil).      

4) In Figure 1 caption, the author should write a sentence about the marked region by dashed square and rectangle.

5) The font size in the Fig. 3 (d) should be corrected.

6) The spacing between the number and unit should be corrected by adding an extra space in the whole manuscript and Figures. Like in the Fig. 3, “20mm” should be corrected “20 mm”.

7) The PVDF and PTFE terms should be define initially.

Author Response

Reviewer #1:

In this manuscript, the authors have prepared a hybrid piezoelectric-electromagnetic-triboelectric generator to harvest energy from the various part of the human body during physical activities. The device structure (design), working principle, and output characteristics, were demonstrated in the manuscript. Additional to the facial structure and non-resonant nature, the hybrid generator is able to effectively harvest the energy at low frequencies from the human body during motions as well as can be utilized for the wearable applications.

The overall evaluation of the current manuscript is good. However, the underlying mechanism and the experimental results of the hybrid energy harvester are little puzzling and unclear. The authors are required to provide further explanations with supporting experimental data. Unless all of the major/minor comments raised by the reviewer is addressed clearly with additional data, this manuscript should not be accepted to the publication in the Journal of Nanomaterials.

We thank reviewer for the careful review and valuable comments. We have revised the entire manuscript carefully according to reviewer’s comments. Please be noted that the revised portions are marked in red in the revised manuscript.

1) In the working principle of EMG, the main influencing factor is the varying magnetic flux, which is dependent on the length of the permanent magnet and number of coils (length of the number of coils). The authors stated about the length of the magnetic bars. However, the length of the coils is not clear; only they mentioned the numbers of turns are 1000. The authors need to put the exact length of the coil.

Since the cavity of the device is made of 3D printing, the size is mainly based on ergonomic considerations, so the inner diameter of the supporting shell is set to 17 mm, the outer diameter is set to 36 mm, and the height is set to 100 mm, and then based on the overall device dimensions and the length of magnets, the length of the groove for winding the magnetic induction coil is determined to be 20 mm, and the depth is determined to be 5 mm. We have also included the information in the revised manuscript.

2) The authors have mentioned that the 30 mm longer magnetic bar produces a strong magnetic flux resulting into higher output voltage and current. While for higher lengths (40, 50, and 60 mm) the magnetic field overlapped leading to lower outcome of the voltage and current. This point need more clear explanation. Why the 30 mm magnetic produces a higher magnetic flux compare to others? The authors did not mentioned about the strength of the magnetic bars. Does the author have any data for the magnetic strength of the each magnetic bar?

We are sorry for the confusion to the reviewer. What we actually mean is that the 30 mm long magnet can produce a higher magnetic flux difference (dФ) during its reciprocating motion. All the magnets are from the same manufacturer of the same type, just with length variations. Thus the actual magnetic field will increase with the length of the magnet. Now we consider two representative positions of the magnet, that is, the left end (right end is the same due to symmetry) and the middle of the cavity, since the difference of magnetic flux in these two positions directly determine the dФ. As shown in Figure R1, when the magnet is at the left end, the magnetic flux through the coil is Ф1, and when the magnet is at the middle, the magnetic flux is Ф2. First, for Ф1, with a smaller magnet length, the produced magnetic flux will be small. Meanwhile, the smaller the magnet is, it will be farther away from the coil. Therefore, the magnetic flux Ф1 for smaller length magnet will be smaller. In other words, Ф1 increases with magnet length. Second, for Ф2 when the magnet is at middle, a longer magnet will produce a stronger magnetic flux, meaning that Ф2 also increases with magnet length. When the magnet length is too small, Ф2 will be very small, thus dФ = Ф2 – Ф1 will also be very small. While when the magnet length is too large, especially when the magnet overlaps with the coil, Ф1 will be very large approaching the magnitude of Ф2, thus in this case the overall dФ = Ф2 – Ф1 will also be very small, although Ф2 can be quite large. Therefore, due to the tradeoff of Ф1 and Ф2, there is an optimal length that results in the strongest magnetic flux difference dФ. Then according to Faraday’s law of induction that output voltage is proportional to dФ, highest output can be obtained at this condition. Based on our measurements, the optimal length for EMG is 30 mm, which can produce the largest dФ during motion and thereby the highest output. We have made the explanation clearer in the revised manuscript.

Figure R1. Magnet position illustration and magnetic flux for small and large length.

3) Does the length of the magnetic bar is smaller, equal, or larger than length of the coil. More importantly, the authors need to compare the length of the coil and magnetic bar. It should be clarified that the overlapping of the magnetic field between the magnet and coil by comparing the length of the coil and magnetic bar (i.e., by taking the length of the magnetic bar smaller, comparable (same size), and longer than the length of the coil).

By ergonomics, we determine the length of the groove for winding the magnetic induction coil to be 20 mm. Meanwhile, based on the above investigation stated in Comment 2, when the length of the magnet is too small, the output will also be small. Thus we investigate the output performance for 5 different magnet lengths – 20, 30, 40, 50, and 60 mm, which are comparable (same size) and longer than the length of the coil. Then based on the measured results in Figure 3 and Figure 4, i.e., the output performance of the TENG unit and EMG unit with respect to different lengths, the optimal magnet is determined for latter practical applications.

Figure 3. Output characteristics of the TENG unit. (a) Triboelectric output voltage corresponding to different stages within one cycle. (b) Output voltages and (c) output currents of the TENG unit under different magnet lengths. (d) Performance comparison of the output voltage and voltage per unit weight for different magnet lengths.

Figure 4. The output characteristics of the EMG unit. (a) The output voltage corresponding to different stages within one cycle. (b) Output voltages and (c) output currents of the EMG unit under different magnet lengths. (d) Performance comparison of the output voltage and voltage per unit weight for different magnet lengths.

4) In Figure 1 caption, the author should write a sentence about the marked region by dashed square and rectangle.

We have illustrated the structure in the dashed box in Figure 1 and marked it in red in the revised manuscript.

5) The font size in the Fig. 3 (d) should be corrected.

We have unified the font size on the images in the revised manuscript.

6) The spacing between the number and unit should be corrected by adding an extra space in the whole manuscript and Figures. Like in the Fig. 3, “20mm” should be corrected “20 mm”.

We have added spaces between all the numbers and units in the revised manuscript.

7) The PVDF and PTFE terms should be define initially.

We have defined Polyvinylidene Fluoride (PVDF) and Polytetrafluoroethylene (PTFE) where they first appeared in the revised manuscript.

Reviewer 2 Report

The paper is very good. The experiment is precise and the results are reliable. I had some small questions but in the course of my reading the authors give very good answers

Author Response

Reviewer #2:

The paper is very good. The experiment is precise and the results are reliable. I had some small questions but in the course of my reading the authors give very good answers.

We thank reviewer for the careful review and supportive comments. We have improved the manuscript further and hope it will be clearer and have higher scientific impacts.

Reviewer 3 Report

Review Comments

The manuscript presents a hybrid nanogenerator made of TENG, EMG, and PENG components and systematically studied the energy harvesting performance of each component. Also, the manuscript demonstrates the practical application of biomechanical energy harvesting by human activity of attaching in human body positions as well as powering a wireless IoT module. There are few minor concerns the authors must address before its publication.

  1. The authors should provide the current response of individual energy harvesting components with respect to various frequency.
  2. The authors must check the power output of EMG under various load matching resistance. As the EMG coil with 70 um diameters, having 1000 turns would have an inbuild resistance value. So how do the authors calculate with different impedance matching resistance?
  3. There are several reports published on the similar configuration of hybrid nanogenerator. Can the authors explain the novelty of the present report compared with the previously published articles.
  4. Some relevant reports would be helpful for the authors and can be cited in the introduction part ACS Appl. Electron. Mater.2020, 2, 10, 3100–3108, Micro and Nano Syst Lett 7, 14 (2019)

Author Response

Reviewer #3:

The manuscript presents a hybrid nanogenerator made of TENG, EMG, and PENG components and systematically studied the energy harvesting performance of each component. Also, the manuscript demonstrates the practical application of biomechanical energy harvesting by human activity of attaching in human body positions as well as powering a wireless IoT module. There are few minor concerns the authors must address before its publication.

We thank reviewer for the careful review and valuable comments. We have revised the entire manuscript carefully according to reviewer’s comments. Please be noted that the revised portions are marked in red in the revised manuscript.

  1. The authors should provide the current response of individual energy harvesting components with respect to various frequency.

The current response of individual energy harvesting components with respect to frequency is tested and included in the revised manuscript, as shown in the blow figure. We can see that when frequency increases, the output voltage and output current of each module increase continuously and the frequency is also accelerated. Due to the limitation of the corresponding equipment, the relationship between the output current and frequency of the friction unit is not measured in this paper. But we can predict that the output current of the corresponding friction module also increases with the increase of frequency. This is because when the frequency increases, the number of work per unit time increases, so the output performance and frequency of each module will increase accordingly.

Figure R.1 Output characteristics of each module at different frequencies (a) voltage output characteristics of TENG modules (b) voltage output characteristics of EMG modules (c) voltage output characteristics of PEG modules (d) current output characteristics of EMG modules (e) current output characteristics of PEG modules

  1. The authors must check the power output of EMG under various load matching resistance. As the EMG coil with 70 um diameters, having 1000 turns would have an inbuild resistance value. So how do the authors calculate with different impedance matching resistance?

In this work, the calculation of the matching resistance of different impedances is obtained by calculating the output power of each module by using different external resistance values. That is, different resistors are connected the EMG unit, and the output voltage is measured. Then based on P = V2/R, the output power is calculated. When the internal resistance of the EMG unit and connected external resistance are the same, maximum output power will be obtained. And the external resistance is the optimal matching resistance. According to the results in Figure 6f, the optimal matching resistance for the EMG unit is 500 Ω.

Figure 6. The output voltages of the TENG unit at (a) different frequencies and (b) different displacements, and (c) the output voltage and power curve under different external loads. The output voltages of the EMG unit at (d) different frequencies and (e) different displacements, and (f) the output voltage and power curve under different external loads. The output voltages of the PEG unit at (g) different frequencies and (h) different displacements, and (i) the output voltage and power curve under different external loads.

  1. There are several reports published on the similar configuration of hybrid nanogenerator. Can the authors explain the novelty of the present report compared with the previously published articles.

Compared with previously reported structures of hybrid nanogenerators, this device has the following advantages:

(1) Simple structure, cheap production cost and easy assembly. The device only needs to be 3D printed with a casing, and then can be readily completed by simple gluing, winding and assembly.

(2) Large frequency response bandwidth and suitable for wearable applications. Due to the non-resonant structural design, the proposed device in our work can respond to a wide range of excitation frequencies, especially low frequencies. Since all the three energy harvesting components are all based on the movements of the magnet, thus their motions are highly synchronized and sound output performance can be obtained. Besides , the device is based on ergonomic design, it can be well applied to various parts of the human body. Just like the experiments we did in this work, it can be conveniently held by hands, placed on arms or legs, for harvesting energy from various human motions.

(3) Unlike the conventional piezoelectric generator that utilizes d33 generation mode, that is, the applied strain is in the same direction with the generated electric field. The PEG unit in our work adopts d31 mode, where the applied strain is in perpendicular direction with the generated electric field. The two end caps are designed to be hollow, as indicated in Figure R2 below, thus the PVDF thin film fixed on the caps will be anchored on both ends, while the middle part is suspended. In this way, when the magnet hits the PVDF thin film, it will deform downward, corresponding to the strain of stretching. Due to the large impact force, the PVDF thin film can have a much higher strain compared to the conventional d33 mode, thus higher output performance will be obtained from this design, which means the human motions can be harvested more effectively.

Figure R2. End cap design.

  1. Some relevant reports would be helpful for the authors and can be cited in the introduction part ACS Appl. Electron. Mater.2020, 2, 10, 3100–3108, Micro and Nano Syst Lett 7, 14 (2019)

Thank you for the suggestion and we have cited these good references in the introduction part of the revised manuscript.

Reviewer 4 Report

The authors present an investigation of a triple hybrid energy harvester, which employs three mechanisms to generate a voltage. The work is well done and the manuscript is well written. However, it might be interesting to compare these results with the use of two devices to see how much of an improvement is obtained with the present design. Please comment.

The authors investigated the effects at a frequency of 4 Hz. It would be useful to see how much the result dependes upon frequency of motion.

Author Response

Reviewer #4:

The authors present an investigation of a triple hybrid energy harvester, which employs three mechanisms to generate a voltage. The work is well done and the manuscript is well written. However, it might be interesting to compare these results with the use of two devices to see how much of an improvement is obtained with the present design. Please comment.

The authors investigated the effects at a frequency of 4 Hz. It would be useful to see how much the result dependes upon frequency of motion.

We thank reviewer for the careful review and valuable comments. We have revised the entire manuscript carefully according to reviewer’s comments. Please be noted that the revised portions are marked in red in the revised manuscript.

As shown in Figure 8b, the capacitor charging performance by different energy harvesting modules and the hybrid device is compared. Based on the charging performance of the 2.2 μF capacitor at the moment of 4 s, the obtained voltage for the EMG, PEG, TENG, and the hybrid device is 11.3 V, 3 V, 0.5 V, and 14.6 V, respectively. Therefore, compared to the individual module, the hybrid device offers an improvement of 29.2 %,386.7 %, and 2820% over the EMG, PEG, and TENG module, respectively.

In this work, corresponding experiments are carried out on the output of each module with different frequencies, as shown in Figure R3 below. We can see that the output voltage of each module of the hybrid device with respect to different frequencies. When the frequency increases, the output performance of each module also increases, showing the output has a positive correlation with frequency. This is because as the frequency increases, the period of the vibrational motion decreases, which means a shorter time for the charge transferring process. Therefore, the output current and the output voltage of all the 3 energy harvesting modules increase with frequency. The test frequency of the follow-up experiments in this work is 4 Hz because this frequency matches the activity frequency of most of the human motions, so that the frequency is set to 4 Hz in the follow-up discussions.

Figure 8. (a) Power management circuit for the hybrid generator. (b) Charging a capacitor of 2.2 μF. (c-e) Powering a wireless IoT module by the hybrid generator.

Figure R3. Output characteristics of each module at different frequencies (a) friction module (b) electromagnetic module (c) piezoelectric module.

Round 2

Reviewer 1 Report

Because the authors responded appropriately to our comments, I recommend that this paper be acceptable to the Nanomaterials journal.